# Quality of Life in Children with Celiac Disease: An Observational Study

**DOI:** 10.3390/nu17061085

**Published:** 2025-03-20

**Authors:** Anna Rozensztrauch, Paulina Mostyńska

**Affiliations:** 1Department of Pediatrics and Coordinated Child Care, Wroclaw Medical University, 50-367 Wroclaw, Poland; 2Division of Fundamentals of Midwifery, Department of Midwifery, Wroclaw Medical University, 50-367 Wroclaw, Poland; paulina.mostyska@umw.edu.pl

**Keywords:** celiac disease, children, quality of life

## Abstract

**Background/Objectives**: Celiac disease (CD) is one of the most common chronic autoimmune disorders affecting children worldwide. The aim is to explore the significance of quality of life (QOL) research in pediatric CD, highlighting the importance of assessing both physical and psychosocial aspects of well-being. **Materials and Methods**: The study used a self-administered questionnaire, which consisted of questions on sociodemographic and clinical characteristics, as well as a general assessment of the QOL by using the validated PedsQL™ 4.0. **Results**: Dietary restrictions were associated with social challenges, as reported by 43% of respondents who indicated their child had experienced exclusion or distress during family gatherings, while 48% encountered difficulties in the school setting. The overall QOL score had a mean of 68.9 (SD = 15.00), with a median of 67.4, and ranged from 41.3 to 100.0, reflecting individual variations in perceived well-being. Age is a significant factor influencing children’s social interactions and experiences within educational settings, likely due to increased academic demands, social pressures, or developmental changes. **Conclusions**: The study suggests that the study factor significantly influences physical functioning and overall quality of life, while its impact on emotional, social, and school domains is comparatively lower.

## 1. Introduction

Celiac disease (CD) is one of the most common chronic autoimmune disorders affecting children worldwide, with varying prevalence rates depending on geographical region, genetic predisposition, and diagnostic practices. The global prevalence of celiac disease is estimated to be around 1% of the population [1]. The seroprevalence of celiac disease is reported at 1.4%, while the biopsy-confirmed prevalence is approximately 0.7% [2]. However, its prevalence varies based on geographical and ethnic differences. The highest rates are observed in Europe and Oceania (both at 0.8%), while the lowest prevalence is found in South America (0.4%) [3,4,5].

In children, the onset of CD typically occurs between the ages of 6 months and 2 years, often coinciding with the introduction of gluten into the diet. However, the disease can develop at any age, and its clinical presentation may vary depending on the child’s age, genetic factors, and environmental exposures [6,7,8]. Early diagnosis is crucial because untreated CD can result in significant health issues, including growth delays, poor weight gain, nutritional deficiencies, and cognitive difficulties, all of which can impact a child’s development and QOL. Epidemiological studies indicate that the incidence of CD is higher in regions with predominantly European ancestry, particularly in countries such as the United States, Canada, and Northern Europe, where genetic risk factors, such as the presence of the HLA-DQ2 or HLA-DQ8 alleles, are more common [9,10,11]. Conversely, the prevalence of CD is lower in regions where these genetic markers are less prevalent, such as parts of Asia and sub-Saharan Africa, though recent studies have indicated a rise in the number of cases in these areas as well, likely due to increased awareness and better diagnostic capabilities [12].

Gender differences in the prevalence of CD are also observed, with a higher incidence in females compared to males. In children, the ratio of affected girls to boys is typically 2:1, which is consistent across many global populations [8]. This discrepancy may be attributed to hormonal differences or other genetic factors that are not yet fully understood. The gender gap narrows somewhat in adulthood, but the overall pattern persists, with females continuing to be more frequently diagnosed with CD than males [13]. The other important issue in the epidemiology of CD in children is the increasing diagnosis of the disease in individuals who do not present with clinical gastrointestinal symptoms. Many children diagnosed with CD may experience only non-specific symptoms, such as fatigue, irritability, or skin rashes, or they may be asymptomatic entirely [14,15]. This has led to a growing number of diagnoses in individuals who might have previously gone undiagnosed, further contributing to the apparent rise in prevalence. Early screening efforts, especially for at-risk populations, such as those with a family history of CD or other autoimmune disorders, have become more routine in many healthcare settings [16,17]. The increasing understanding of CD and its growing recognition in both clinical and research contexts emphasize the need for improved diagnostic methods and more comprehensive population-based studies. These efforts are essential to understand the full scope of the disease, its regional variation, and the potential long-term impact of CD on children’s health. Research indicates that adherence to a strict gluten-free diet is key to preventing health complications, but can pose challenges in daily functioning [18,19].

In addition, as noted by McDermid et al. [20], children with CD often experience stress related to dietary restrictions and social difficulties, highlighting the need for psychological and educational support. Research on the quality of life (QOL) of pediatric patients with CD is important for developing strategies to improve their well-being and facilitate adaptation to the disease. CD is a chronic autoimmune disorder that significantly impacts children’s QOL, both physically and psychosocially. Studies indicate that strict adherence to a gluten-free diet is essential for preventing health complications but can pose daily challenges. Research on the QOL of pediatric celiac patients is crucial for developing strategies to improve their well-being and facilitate adaptation to the disease [21,22]. By understanding the impacts of CD on QOL, it is essential to develop comprehensive care strategies that not only address a child’s medical needs, but also support his or her psychological, social, and emotional well-being. Despite the importance of QOL, research on the topic in children with CD remains limited, and most of the available evidence focuses primarily on clinical outcomes or adult populations. Nevertheless, children present a unique set of challenges and considerations, as their well-being is deeply linked to developmental factors, school life, peer interactions, and family dynamics. Therefore, evaluating QOL in patients with pediatric CD provides important insight into the broader impact of the disease on a child’s life beyond clinical measurements. QOL encompasses a range of physical, emotional, social, and functional dimensions, and for children with CD, these aspects can be significantly altered by the need to adhere to a strict gluten-free diet, the potential for social isolation, and the emotional burden associated with living with a chronic illness [23,24]. The aim is to explore the significance of QOL research in pediatric CD, highlighting the importance of assessing both physical and psychosocial aspects of well-being.

## 2. Materials and Methods

### 2.1. Characteristics of the Study Group

The study included children with confirmed CD, affiliated with the Association of People with Celiac Disease. The study used a self-administered questionnaire, which consisted of questions on sociodemographic and clinical characteristics, as well as a general assessment of the QOL of children with CD. Inclusion criteria for the study: a confirmed diagnosis of celiac disease based on serology and small bowel biopsy, being a member of the Association of People with Celiac Disease, the consent of a legal guardian to participate in the study, and the ability to complete the questionnaire independently or assistance from a guardian in completing it. Participants who met one or more of the following criteria were excluded from the study: no confirmed diagnosis of celiac disease, coexisting chronic diseases that could affect quality of life independent of CD (e.g., cystic fibrosis, type 1 diabetes, cancer), failure to adhere to a gluten-free diet for at least 6 months prior to the study, lack of consent from a legal guardian to participate in the study, and problems with understanding and completing the questionnaire despite assistance from a caregiver.

### 2.2. General Health-Related Quality of Life Instrument

The QOL questionnaires were measured by using the validated PedsQL general questionnaire. This was the Paediatric Quality of Life (PedsQL) 4.0 questionnaire, which comprises 23 items and evaluates QOL over the past month in preschool-aged children (ages 5–7), school-aged children (ages 8–12), and adolescents (ages 13–18). For children aged 2–4 years, the report comprises 21 items and does not involve scales of school functioning and communication. Respondents score the items on a 5-point Likert scale, where 0 means “never” and 4 means “almost always”. The questionnaire items are reverse-scored and linearly converted to a 0–100 scale (0 = 100, 1 = 75, 2 = 50, 3 = 25, 4 = 0), where 100 represents the best quality of life. The overall score is the sum of average scores from each subscale. The lower the score, the lower the QOL [25,26,27,28].

### 2.3. Statistical Analysis

Quantitative variables, which are expressed numerically, were analyzed by calculating the mean, standard deviation, median, quartiles, as well as minimum and maximum values. For qualitative variables, which cannot be expressed numerically, the frequency and percentage of each category were determined. Fisher’s exact test was used to compare qualitative variables, while quantitative variables were analyzed using Student’s *t*-test for normally distributed data and the Mann–Whitney–Wilcoxon U test for non-normally distributed data. When comparing quantitative variables across three groups, analysis of variance (ANOVA) was applied. Correlations between quantitative variables were assessed using Pearson’s correlation coefficient for normally distributed data and Spearman’s correlation coefficient otherwise. The Shapiro–Wilk test was employed to evaluate the normality of quantitative variables, and Levene’s test was used to assess variance homogeneity. A significance threshold of 0.05 was set, meaning p-values below this level were considered statistically significant. All calculations were carried out using R software, version 3.6.1.

## 3. Results

### 3.1. Characteristics of the Study Population and Diagnostic Procedures for Celiac Disease

A total of 129 respondents participated in the study, with women comprising the majority (93%) and men accounting for 7%. The mean age of parents was 40.0 years (SD = 6.53). Most respondents (81%) resided in urban areas, with 19% living in rural regions. The majority had attained higher education (69%), while 26% had secondary education. Most families had two children (58%), followed by one (16%), three (14%), or four children (12%). The mean maternal age at childbirth was 30.4 years (SD = 5.17), with the youngest mother being 19 and the oldest 45 years. Vaginal delivery was reported in 59% of cases, whereas 41% of children were born via cesarean section. Most children were born at term (74%), while 26% were delivered after 38 weeks of gestation. Immediate postnatal skin-to-skin contact through breastfeeding was reported in 81% of cases. The study population was composed of 52% female and 48% male children, with a mean age of 9.3 years (SD = 4.21), ranging from 2 to 18 years.

Serological testing for celiac-specific antibodies was the most frequently performed diagnostic procedure (91%), followed by a histopathological examination of biopsy samples obtained during gastroscopy (54%). Genetic testing for HLA DQ2 and DQ8 alleles was conducted in 20% of cases. Prior to celiac disease diagnosis, the most prevalent symptom was abdominal pain (64%), followed by chronic fatigue and drowsiness (36%), abdominal bloating (36%), steatorrhea or watery diarrhea (35%), weight loss and malnutrition (33%), constipation (32%), anemia (31%), short stature (30%), and loss of appetite (23%). Less commonly reported symptoms included musculoskeletal pain (20%), aphthous ulcers (19%), low muscle mass (17%), developmental delays (16%), atopic dermatitis (16%), vomiting (13%), autoimmune diseases (12%), brain fog (10%), eating disorders (10%), and hair loss (10%).

The most frequently reported symptom following gluten ingestion was abdominal pain (68%), followed by steatorrhea or watery diarrhea (31%), abdominal bloating (23%), weight loss and malnutrition (11%), aphthous ulcers (9%), hair loss (8%), dermatitis herpetiformis or other dermatological symptoms (7%), loss of appetite (5%), migraine headaches (5%), and atopic dermatitis (5%). Less common manifestations included developmental delays (3%), anemia (3%), musculoskeletal pain (3%), short stature (3%), dizziness (2%), brain fog (2%), depression (2%), eating disorders (2%), and gluten ataxia (1%). Notably, 16% of parents reported no observable symptoms in their child following gluten consumption.

The mean duration of adherence to a gluten-free diet was 3.2 years (SD = 3.79), with a range from 1 month to 18 years. The interquartile distribution indicated that 25% of children maintained the diet for up to 6 months, 50% for up to 2 years, and 75% for up to 4 years. The most frequently reported duration was 3 years. Strict adherence to a gluten-free diet was reported in 93% of cases, while 5% of children occasionally consumed gluten, and 2% did not adhere to dietary restrictions. Parental strategies for managing accidental gluten exposure primarily involved passive monitoring until symptom resolution (15%) or the administration of probiotics (15%). Analgesics and antispasmodics were used in 7% of cases, while prokinetic agents and warm compresses were each applied in 5% of cases.

Following the initiation of a gluten-free diet, abdominal pain resolved 61% of children. Additional improvements were observed in cases of watery or steatorrheic diarrhea (33%), weight loss (28%), abdominal bloating (19%), loss of appetite (17%), anemia (16%), aphthous ulcers (12%), short stature (10%), drowsiness (9%), atopic dermatitis (8%), mood disturbances (8%), hair loss (8%), skin conditions (7%), constipation (7%), musculoskeletal pain (5%), dizziness (5%), brain fog (5%), low muscle mass (5%), developmental delays (3%), depression (2%), early-onset osteoporosis (2%), arthritis (2%), elevated transaminase levels (1%), and recurrent fractures (1%). However, 7% of parents reported no symptom resolution following dietary modification.

In most cases (72%), only the child had a celiac disease diagnosis, whereas the condition was also reported in the mother (14%), a sibling (12%), a maternal aunt (5%), a maternal grandmother (2%), a son (2%), a daughter (2%), or a father (2%). Most children (65%) had no coexisting conditions. Health improvements were most frequently observed within the first month of adhering to the gluten-free diet (33%).

Dietary restrictions were associated with social challenges, as reported by 43% of respondents who indicated their child had experienced exclusion or distress during family gatherings, while 48% encountered difficulties in the school setting. A small percentage of children were excluded from school trips due to their dietary requirements.

### 3.2. Quality of Life Assessment in Children with Celiac Disease

#### 3.2.1. Descriptive Statistics of PedsQL Scores

Table 1 presents the descriptive statistics of QOL scores across various functional domains. The highest mean score was observed in the physical functioning (PF) domain (M = 77.2, SD = 16.57), indicating that children generally reported better physical well-being compared to other domains. The lowest mean score was recorded in the functioning in daycare/preschool/school (SCHF) domain (M = 64.0, SD = 18.87), suggesting that children faced more challenges in educational settings.

The overall QOL score had a mean of 68.9 (SD = 15.00), with a median of 67.4, and ranged from 41.3 to 100.0, reflecting individual variations in perceived well-being.

The SF domain exhibited a broad range of scores (30.0–100.0), with a median of 65.0, highlighting substantial differences in children’s ability to engage in social interactions. The EF domain had a mean score of 58.7 (SD = 17.87), suggesting a relatively lower perceived emotional well-being compared to physical and social domains.

#### 3.2.2. Comparison of PedsQL Scores by Child’s Gender

Table 2 presents the comparative analysis of QOL scores by genders evaluated using the Mann–Whitney U test. The analysis revealed no statistically significant differences between genders across all assessed domains, as all *p*-values exceeded the standard significance threshold (*p* > 0.05). The PF domain showed slightly higher mean scores among boys (M = 79.1, SD = 16.87) compared to girls (M = 75.5, SD = 16.22), with a median of 87.5 for boys and 75.0 for girls. Despite this difference, the result was not statistically significant (Z = −1.83, *p* = 0.067). In the EF domain, the mean scores for girls (M = 59.0, SD = 16.97) and boys (M = 58.4, SD = 18.92) were nearly identical, both with a median of 55.0, indicating no gender-related differences (Z = −0.01, *p* = 0.992). Similarly, the SF domain showed comparable scores between girls (M = 71.3, SD = 22.79) and boys (M = 68.9, SD = 20.87), with no significant difference (Z = −0.34, *p* = 0.733). The SCHF domain showed a slight advantage for boys (M = 66.8, SD = 16.29) over girls (M = 61.3, SD = 20.76), but this difference did not reach statistical significance (Z = −0.77, *p* = 0.439). Finally, the QOL scores were similar between the two groups (M = 68.2, SD = 15.58 for girls and M = 69.7, SD = 14.43 for boys), with medians of 66.3 and 67.7, respectively. The Mann–Whitney U test confirmed no significant gender differences (Z = −0.37, *p* = 0.715).

The findings indicate that there are no substantial differences between boys and girls in terms of their perceived QOL across various domains, such as physical, emotional, social, and school-related aspects. This suggests that both genders experience similar levels of well-being and daily functioning, meaning that gender does not appear to be a significant factor in shaping their overall QOL.

#### 3.2.3. Correlation Between Child’s Age and PedsQL Scores

Spearman’s rank correlation analysis was conducted to evaluate the relationship between child’s age and various domains of the QOL. The results, presented in Table 3, indicate several significant negative correlations, suggesting that older children tend to report lower quality of life in specific domains. A statistically significant negative correlation was found between age and social functioning (ρ = −0.28, *p* = 0.001), indicating that as children grow older, they experience greater difficulties in social interactions. Additionally, a strong negative correlation was observed between age and functioning in daycare/preschool/school (ρ = −0.38, *p* < 0.001), suggesting that older children tend to perceive greater challenges in their educational or childcare environments. The overall QOL score also showed a significant negative correlation with age (ρ = −0.27, *p* = 0.002), implying a decline in perceived well-being as children grow older. However, no statistically significant correlations were observed for PF (ρ = −0.11, *p* = 0.201) or EF (ρ = −0.16, *p* = 0.072).

The findings indicate that age is a significant factor influencing children’s social interactions and experiences within educational settings, likely due to increased academic demands, social pressures, or developmental changes. However, the lack of significant correlations in physical and emotional functioning suggests that these aspects of well-being are less affected by age-related factors during early childhood.

#### 3.2.4. Influence of Place of Residence on PedsQL Scores

Spearman’s correlation coefficients (ρ) and corresponding *p*-values for the PedsQL domains are as follows:PF: A moderate positive correlation (ρ = 0.48) was observed, with a highly significant *p*-value (<0.001), indicating a strong relationship between the variable of interest and physical functioning. This suggests that as the related factor increases, physical functioning also tends to improve.EF: The correlation for emotional functioning is negligible (ρ = 0.02), with a *p*-value of 0.864, which is not statistically significant. This suggests that there is no meaningful relationship between the variable of interest and emotional functioning in this sample.SF: The correlation coefficient for social functioning is effectively zero (ρ = 0.00), and the *p*-value is 0.961, indicating no significant relationship between the variable of interest and social functioning. This implies that social functioning is unaffected by the factor under consideration.SCHF: A low positive correlation (ρ = 0.17) was found, with a *p*-value of 0.049, which is statistically significant at the 0.05 level. While the correlation is weak, it indicates a slight association between the variable of interest and school-related functioning.Overall QOL: A moderate positive correlation (ρ = 0.26) was observed, with a statistically significant *p*-value of 0.003. This suggests a moderate relationship between the variable of interest and overall quality of life, with an increase in the factor linked to an improvement in the overall quality of life.

These findings suggest that while some domains, particularly physical functioning and overall QOL, show moderate correlations, others like emotional and social functioning appear to be unaffected by the factor examined (Table 4).

Table 5 indicates the quality of life of children in the control group.

## 4. Discussion

To the best of our knowledge, this is the first prospective long-term follow-up study on HRQoL on CD. Our study provides valuable insights into the impact of CD on the QOL of children and adolescents in Poland. An analysis of the results allows for a better understanding of the factors affecting patients’ well-being and the possibilities of improving their functioning in daily life. The analysis of our findings offers a deeper understanding of the factors influencing patients’ well-being and highlights opportunities for improving their daily functioning. The results of this study provide a comprehensive overview of the QOL scores across different functional domains in children with CD. The highest mean score observed in the physical functioning suggests that children generally reported better physical well-being compared to other domains. This aligns with existing research, where physical health and functioning tend to be relatively stable and easier to assess, reflecting fewer barriers to engagement in physical activities compared to emotional or social challenges [29]. The children in this study appeared to have relatively positive perceptions of their physical abilities, possibly due to the favorable management of their conditions, such as proper medical care or effective coping strategies. Children with chronic diseases, including CD, face significant difficulties in school functioning, as evidenced in the scientific literature. The most common challenges include increased absenteeism, difficulty concentrating, and feelings of isolation due to the need to follow a restrictive diet [30]. School plays a key role in cognitive, social, and emotional development, so difficulties in this area can have long-term consequences for psychological well-being and academic achievement. Barriers present in the educational environment call for the implementation of effective, evidence-based interventions and the adaptation of school settings to make them more inclusive and supportive of children with CD. As observed in earlier studies, children’s perceptions of QOL vary considerably, likely influenced by factors such as disease severity, access to healthcare, the availability of social support, and psychological coping mechanisms. This significant heterogeneity underscores the multidimensional nature of QOL and emphasizes the need for a personalized approach when assessing well-being in this population. A holistic approach to understanding these individual differences is essential for developing targeted interventions to improve overall health outcomes. The observed variability in the social functioning of children with chronic diseases emphasizes the complex interaction of the factors influencing their ability to involve in social interactions. Social challenges are especially clear in the presence of dietary restrictions, as is the case with CD, where worries about food choices can lead to social isolation or increased anxiety [31]. Differences in social functioning may also stem from variations in family dynamics, social environments, and the level of support and understanding from peers. These findings underscore the need for targeted interventions that promote social integration and address the psychosocial impact of dietary restrictions in this population. The emotional functioning seems to be comparatively lower than the physical and social domains among children with chronic conditions, underscoring the significant emotional burden of coping with these diseases. Diseases such as CD can contribute to emotional distress, particularly due to the social challenges of adhering to dietary restrictions, frequent medical visits, and feeling different from peers [16]. This psychological burden highlights the need for complex care management strategies that address not only the physical aspects of chronic illness, but also the emotional well-being of affected children. Integrating psychological support into routine care can help alleviate these challenges and improve overall QOL. No significantly different results were noted in any of the PedsQL domains, suggesting that boys and girls with CD report similar experience of physical, emotional, social, and school well-being. These findings are also consistent with other studies on QOL in children with chronic diseases, including CD. Studies have found minimal differences in overall health perceptions among children with chronic diseases by gender. Additionally, research indicates that in children with CD, factors such as adherence to a gluten-free diet, disease management, and family support have a more significant impact on their QOL than gender alone. The absence of gender differences in QOL scores in our study may reflect the predominant influence of these factors over gender [32].

Furthermore, research indicates that dietary non-adherence and socioeconomic challenges related to maintaining a gluten-free diet are primary contributors to diminished health-related quality of life in both children and adults with celiac disease. This underscores the importance of comprehensive support systems and accessible resources to assist patients and their families in managing the disease effectively.

Studies have found minimal differences in overall health perceptions among children with chronic diseases by gender. Additionally, research indicates that in children with CD, factors such as adherence to a gluten-free diet, disease management, and family support have a more significant impact on their QOL than gender alone. The absence of gender differences in QOL scores in our study may reflect the predominant influence of these factors over gender. Age is a significant determinant in children’s social and educational experiences. The negative correlations between age and both social functioning (ρ = −0.28, *p* = 0.001) and school-related functioning (ρ = −0.38, *p* < 0.001) suggest that older children report more difficulties in these domains. This may be attributed to increased academic demands, developmental changes, and escalating social pressures that accompany aging. As children mature, they encounter more complex social dynamics and academic challenges, potentially impacting their overall QOL. The negative correlation with overall QOL (ρ = −0.27, *p* = 0.002) supports the notion that older children may experience a decline in perceived well-being as they face these challenges. These findings underscore the necessity for age-specific interventions aimed at supporting social and educational functioning in children with celiac disease. Tailored strategies that address the unique challenges faced by older children, such as peer relationship management and coping mechanisms for academic stress, could be beneficial. Moreover, fostering a supportive environment both at home and in educational settings is crucial to mitigate the adverse effects associated with aging in this population. In conclusion, while gender does not appear to significantly influence QOL in children with celiac disease, age-related factors play a critical role. Addressing these age-specific challenges through targeted interventions may enhance the overall well-being and daily functioning of these children [33,34,35].

It is worth mentioning that the finding that older children report lower scores in social and school functioning requires further elaboration. During adolescence, increased peer pressure and greater awareness of being different due to a chronic condition like CD may lead to feelings of isolation and stigma [36]. Additionally, growing academic demands can pose an extra burden for children adhering to a strict gluten-free diet, potentially affecting their daily school performance [37,38]. Understanding these factors is crucial for developing effective psychosocial support strategies for children with CD.

Despite the strengths of our study, several limitations should be acknowledged. First, the study was conducted in a single-center setting, which may limit the generalizability of our findings to broader populations. Future multi-center studies are needed to confirm our results in diverse clinical and sociodemographic contexts. Second, the reliance on self-reported questionnaires for assessing health-related quality of life may introduce response bias. Participants’ subjective perceptions of their well-being could be influenced by factors such as mood, parental influence, or recall bias. Objective measures or clinician-assessed tools could complement self-reported data to provide a more comprehensive understanding. Third, our study did not account for potential confounding variables such as socioeconomic status, parental education, and access to specialized dietary resources, which could influence QOL outcomes [39]. Future research should incorporate these factors to better understand their impact on QOL in children with CD. Lastly, while our study assessed QOL at a single time point, a longitudinal approach would provide deeper insight into how QOL evolves over time in response to disease management, dietary adherence, and psychosocial factors. Future studies should consider long-term follow-ups to identify trends and potential intervention points.

A final major limitation of this study is the use of a generic QOL assessment tool, the PedsQL, rather than a disease-specific instrument tailored to CD. Although the PedsQL enables broad comparisons among pediatric populations and includes both physical and psychosocial aspects of well-being, it may not fully reflect the unique struggles associated with CD. Disease-specific questionnaires, such as the CD-QOL, have been validated to assess QOL in patients with CD and may provide a more accurate assessment of disease load. This approach would allow for a direct comparison of their respective domains, offering a more comprehensive understanding of the impact of CD on children’s QOL. In addition, future research should explore how disease-specific and general measures of QOL complement each other in assessing both disease-specific and broader psychosocial aspects of well-being in pediatric celiac patients.

Despite these limitations, our findings contribute valuable knowledge to the understanding of quality of life in children with celiac disease and highlight the need for targeted support strategies.

## 5. Conclusions

This study implies that the study factor has a considerable impact on physical functioning and overall QOL, with lower association in the emotional, social and school domains. These findings are consistent with previous research on chronic diseases, particularly CD, where physical health is often most directly dependent on disease management.

## Figures and Tables

**Table 1 nutrients-17-01085-t001:** Descriptive statistics of PedsQL scores.

PedsQL	M	SD	Q1	Me	Q3	Mo	Min.	Max.
PF	77.2	16.57	62.5	81.3	90.6	87.5	43.8	100.0
SF	70.1	21.83	50.0	65.0	95.0	100.0	30.0	100.0
SCHF	64.0	18.87	50.0	60.0	75.0	50.0	16.7	100.0
EF	58.7	17.87	45.0	55.0	70.0	50.0	20.0	100.0
Overall QOL	68.9	15.00	57.1	67.4	82.3	50.0	41.3	100.0

M—mean; SD—standard deviation; Q1—first quartile; Me—median; Q3—third quartile; Mo—mode; Min.—minimum value; Max.—maximum value; PF—physical functioning; SF—social functioning; SCHF—daycare/preschool/school functioning; EF—emotional functioning.

**Table 2 nutrients-17-01085-t002:** PedsQL scores by child’s gender—Mann–Whitney U test results.

PedsQL	Girls (M ± SD)	Me	Boys (M ± SD)	Me	Z	*p*-Value
PF	75.5 ± 16.22	75.0	79.1 ± 16.87	87.5	−1.83	0.067
EF	59.0 ± 16.97	55.0	58.4 ± 18.92	55.0	−0.01	0.992
SF	71.3 ± 22.79	70.0	68.9 ± 20.87	60.0	−0.34	0.733
SCHF	61.3 ± 20.76	50.0	66.8 ± 16.29	65.0	−0.77	0.439
Overall QOL	68.2 ± 15.58	66.3	69.7 ± 14.43	67.7	−0.37	0.715

M—mean; SD—standard deviation; PF—physical functioning; SF—social functioning; SCHF—daycare/preschool/school functioning; EF—emotional functioning; *p*-value (0.005).

**Table 3 nutrients-17-01085-t003:** Spearman’s correlation between child’s age and PedsQL domains.

PedsQL	ρ	*p*-Value
PF	−0.11	0.201
EF	−0.16	0.072
SF	−0.28	0.001
SCHF	−0.38	<0.001
Overall QOL	−0.27	0.002

PF—physical functioning; SF—social functioning; SCHF—daycare/preschool/school functioning; EF—emotional functioning; *p*-value (0.005); ρ—Spearman’s correlation coefficient.

**Table 4 nutrients-17-01085-t004:** Spearman’s rank correlation between place of residence and PedsQL scores.

PedsQL	ρ	*p*-Value
PF	0.48	<0.001
EF	0.02	0.864
SF	0.00	0.961
SCHF	0.17	0.049
Overall QOL	0.26	0.003

PF—physical functioning; SF—social functioning; SCHF—daycare/preschool/school functioning; EF—emotional functioning; *p*-value (0.005); ρ—Spearman’s correlation coefficient.

**Table 5 nutrients-17-01085-t005:** QOL of children with CD vs. control group [24].

PedsQL Score	*n*	Me	SD	*n*	Me	SD
		Study Group			Control Group	
Total score	129	68.90	15.00	6530	80.40	16.10
Physical functioning	129	72.20	16.57	6519	82.11	20.63
Emotional functioning	129	58.70	27.87	6517	80.00	17.25
Social functioning-children	129	64.00	18.87	6533	79.49	15.87

Me—median; SD—standard deviation.

## Data Availability

The data that support the findings of this study are available from the corresponding author, upon reasonable request due to privacy.

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
