# Peer review of "Quality of Life in Children with Celiac Disease: An Observational Study"

_nutrients, 2025, doi:10.3390/nu17061085_

Round 1
Reviewer 1 Report
Comments and Suggestions for Authors
This study addresses a significant topic by exploring the quality of life (QOL) in children with celiac disease (CD), with a focus on both physical and psychosocial aspects. However, several critical issues related to study design, methodology, clarity, and the depth of discussion must be addressed before the manuscript is ready for publication.
- The study’s objective is not clearly defined and appears inconsistently stated between the abstract and introduction.
- The use of the PedsQL general questionnaire is not sufficiently justified. Given that a disease-specific instrument, such as the CD-QOL, has been validated for assessing QOL in CD patients, the rationale for choosing a general measure over a more targeted tool needs to be explained. Additionally, the limitations of using a generic QOL assessment rather than one tailored to CD should be discussed.
- The study presents QOL scores in children with CD but lacks a comparison with a control group. Without a reference population of healthy children, it is difficult to determine whether the observed QOL impairments are unique to CD or reflective of general chronic disease burden. If possible, incorporating a control group would strengthen the findings. If retrospective data is unavailable, the authors should compare their results with existing literature on QOL in pediatric chronic diseases.
- Methodological details are not clearly outlined. The criteria for sample selection should be explicitly stated, along with an explanation of how the sample size was determined. Additionally, it remains unclear whether confounding variables such as socioeconomic status, family support, and disease duration were accounted for. A detailed description of the statistical tests used to analyze group differences is also necessary.
- The finding that older children report lower scores in social and school functioning requires further elaboration. Potential factors such as increased peer pressure, social stigma, and academic demands during adolescence should be explored to provide a more comprehensive interpretation.
- Some epidemiological data are outdated and should be supplemented with more recent references to ensure the study reflects the latest research developments.
Addressing these concerns will enhance the study’s clarity, methodological rigor, and overall contribution to the field.
Author Response
Dear Reviewer, 1
Thank you very much for sending us the consensus opinion about the requested revision of our manuscript entitled: Quality of Life in Children with Celiac Disease: An Observational Study. We appreciate the thoughtful comments, and we have modified the manuscript in response to your suggestions, which we believe will further improve its quality.
REVIEWER COMMENTS 1
This study addresses a significant topic by exploring the quality of life (QOL) in children with celiac disease (CD), with a focus on both physical and psychosocial aspects. However, several critical issues related to study design, methodology, clarity, and the depth of discussion must be addressed before the manuscript is ready for publication.
Thank you for these comments.
REVIEWER COMMENTS 2
The study’s objective is not clearly defined and appears inconsistently stated between the abstract and introduction.
Please see the following revisions to be described in the identical way in the abstract as in the introduction
..”The aim is to explore the significance of quality of life (QOL) research in pediatric CD, highlighting the importance of assessing both physical and psychosocial aspects of well-being…”
REVIEWER COMMENTS 3
The use of the PedsQL general questionnaire is not sufficiently justified. Given that a disease-specific instrument, such as the CD-QOL, has been validated for assessing QOL in CD patients, the rationale for choosing a general measure over a more targeted tool needs to be explained. Additionally, the limitations of using a generic QOL assessment rather than one tailored to CD should be discussed.
Thank you very much for your valuable feedback. The decision to use the PedsQL general questionnaire was based on its broad applicability in assessing overall quality of life across multiple domains, enabling comparisons with other pediatric populations. While disease-specific instruments like the CD-QOL provide a more targeted evaluation, they are primarily designed for adults and may not fully capture the unique challenges faced by pediatric patients. Additionally, using a generic tool allows us to assess both disease-related and broader psychosocial impacts, which are crucial in understanding the overall well-being of children with CD. However, we acknowledge the limitations of this approach and will address them in the discussion. Furthermore, we are considering conducting a follow-up study that incorporates the CD-QOL questionnaire alongside the PedsQoL. This would allow for a direct comparison of their respective domains and provide a more comprehensive evaluation of quality of life in pediatric CD patients.
Please see incorporated the following paragraph in limitation section in discussion.
..”A final major limitation of this study is the use of a generic QOL assessment tool, the PedsQL, rather than a disease-specific instrument tailored to CD. Although the PedsQoL enables broad comparisons among pediatric populations and includes both physical and psychosocial aspects of well-being, it may not fully reflect the unique struggles associated with CD. Disease-specific questionnaires, such as the CD-QOL, have been validated to assess QOL in patients with CD and may provide a more accurate assessment of disease load. This approach would allow a direct comparison of their respective domains, offering a more comprehensive understanding of the impact of CD on children's QOL. In addition, future research should explore how disease-specific and general measures of QOL complement each other in assessing both disease-specific and broader psychosocial aspects of well-being in pediatric celiac patients…”
REVIEWER COMMENTS 4
The study presents QOL scores in children with CD but lacks a comparison with a control group. Without a reference population of healthy children, it is difficult to determine whether the observed QOL impairments are unique to CD or reflective of general chronic disease burden. If possible, incorporating a control group would strengthen the findings. If retrospective data is unavailable, the authors should compare their results with existing literature on QOL in pediatric chronic diseases.
Thank you for your valuable comments and suggestions. We acknowledge the importance of including a control group to better assess whether the observed impairments in quality of life (QOL) are specific to celiac disease or part of a broader chronic disease burden. While retrospective data for a control group was not available, we have addressed this limitation by incorporating comparisons with existing literature on QOL in pediatric chronic diseases. Additionally, we have added Table 5 to provide further context and strengthen our findings. We appreciate your insightful feedback, which has helped us improve the clarity and robustness of our study.
Thank you for this comment. After reviewing suggestion, we have incorporated new table 3 in the results section.
PedsQL score |
n |
Me |
SD |
n |
Me |
SD |
|
|
Study group |
|
|
Control group |
|
Total score |
129 |
68.90 |
15.00 |
6530 |
80.40 |
16.10 |
Physical functioning |
129 |
72.20 |
16.57 |
6519 |
82.11 |
20.63 |
Emotional functioning |
129 |
58.70 |
27.87 |
6517 |
80,00 |
17.25 |
Social functioning-children |
129 |
64.00 |
18.87 |
6533 |
79,49 |
15.87 |
Table 5. QOL in CD and control group.
M—mean; SD—standard deviation.
REVIEWER COMMENTS 5
Methodological details are not clearly outlined. The criteria for sample selection should be explicitly stated, along with an explanation of how the sample size was determined. Additionally, it remains unclear whether confounding variables such as socioeconomic status, family support, and disease duration were accounted for. A detailed description of the statistical tests used to analyze group differences is also necessary.
Quantitative variables, which are expressed numerically, were analyzed by calculating the mean, standard deviation, median, quartiles, as well as minimum and maximum values. For qualitative variables, which cannot be expressed numerically, the frequency and percentage of each category were determined. Fisher’s exact test was used to compare qualitative variables, while quantitative variables were analyzed using the Student’s t-test for normally distributed data and the Mann–Whitney–Wilcoxon U test for non-normally distributed data. When comparing quantitative variables across three groups, analysis of variance (ANOVA) was applied. Correlations between quantitative variables were assessed using Pearson’s correlation coefficient for normally distributed data and Spearman’s correlation coefficient otherwise. The Shapiro–Wilk test was employed to evaluate the normality of quantitative variables, and Levene’s test was used to assess variance homogeneity. A significance threshold of 0.05 was set, meaning p-values below this level were considered statistically significant. All calculations were carried out using R software, version 3.6.1.
REVIEWER COMMENTS 6
The finding that older children report lower scores in social and school functioning requires further elaboration. Potential factors such as increased peer pressure, social stigma, and academic demands during adolescence should be explored to provide a more comprehensive interpretation.
Thank you for this comments, please see incorporated the following sentences in discussion.
..”It is worth mentioning that the finding that older children report lower scores in social and school functioning requires further elaboration. During adolescence, increased peer pressure and greater awareness of being different due to a chronic condition like CD may lead to feelings of isolation and stigma {37]. Additionally, growing academic demands can pose an extra burden for children adhering to a strict gluten-free diet, potentially affecting their daily school performance [38]. Understanding these factors is crucial for developing effective psychosocial support strategies for children with CD. ..”
REVIEWER COMMENTS 7
Some epidemiological data are outdated and should be supplemented with more recent references to ensure the study reflects the latest research developments.
Thank you very much for this epidemiological data, we reflect the latest research and incorporated in the introduction.
…”The global prevalence of celiac disease is estimated to be around 1% of the population [1]. The seroprevalence of celiac disease is reported at 1.4%, while the biopsy-confirmed prevalence is approximately 0.7% [2]. However, its prevalence varies based on geographical and ethnic differences. The highest rates are observed in Europe and Oceania (both at 0.8%), while the lowest prevalence is found in South America (0.4%)[3-5]…”
Addressing these concerns will enhance the study’s clarity, methodological rigor, and overall contribution to the field.
Thank you very much for your valuable comments and all your efforts in conducting this review.

Reviewer 2 Report
Comments and Suggestions for Authors
Very interesting article whose aim is to take stock of celiac disease in pediatric age and its influence on the quality of life through the administration of a questionnaire that allows them to quantify the disorders and measure how much this disabling pathology can interfere with daily activities. The framework given by colleagues in the introduction is very good, I would add that often it is the parents who, noticing the quality of the stool and the tendency of the child not to grow, take charge of requesting the intervention of the pediatrician who will then set in motion all the diagnostics to reach the diagnosis. We must add that the pathology, rather widespread, has meant that especially in second but especially third level hospitals there is a nutritional team that mainly deals with nutritional problems of oncology patients (doi.org/10.3390/nu17010188 to read and cite in bibliography), but generally also takes charge of the problems of the pediatric world. In Western countries, all schools, restaurants and other centers that host celiac patients are now equipped to serve gluten-free meals. The possibility of administering a questionnaire is a useful system for measuring the quality of life, the only problem is who administers the questionnaire, parents and/or the caregiver could influence the response, children are often not able to give autonomous answers, a third person, identified in a social worker of the reference hospital could perhaps be more suitable. The discussion and conclusions of colleagues are in line with the rest of the paper and the bibliography supports all the reasoning of the case, we agree with the limitations. The English is good, the bibliography equally
Comments on the Quality of English Languagegood english
Author Response
Dear Reviewer, 2
Thank you very much for sending us the consensus opinion about the requested revision of our manuscript entitled: Quality of Life in Children with Celiac Disease: An Observational Study. We appreciate the thoughtful comments, and we have modified the manuscript in response to your suggestions, which we believe will further improve its quality.
REVIEWER COMMENTS 1
Very interesting article whose aim is to take stock of celiac disease in pediatric age and its influence on the quality of life through the administration of a questionnaire that allows them to quantify the disorders and measure how much this disabling pathology can interfere with daily activities. The framework given by colleagues in the introduction is very good, I would add that often it is the parents who, noticing the quality of the stool and the tendency of the child not to grow, take charge of requesting the intervention of the pediatrician who will then set in motion all the diagnostics to reach the diagnosis. We must add that the pathology, rather widespread, has meant that especially in second but especially third level hospitals there is a nutritional team that mainly deals with nutritional problems of oncology patients (doi.org/10.3390/nu17010188 to read and cite in bibliography), but generally also takes charge of the problems of the pediatric world. In Western countries, all schools, restaurants and other centers that host celiac patients are now equipped to serve gluten-free meals. The possibility of administering a questionnaire is a useful system for measuring the quality of life, the only problem is who administers the questionnaire, parents and/or the caregiver could influence the response, children are often not able to give autonomous answers, a third person, identified in a social worker of the reference hospital could perhaps be more suitable. The discussion and conclusions of colleagues are in line with the rest of the paper and the bibliography supports all the reasoning of the case, we agree with the limitations. The English is good, the bibliography equally.
Thank you for your insightful and constructive feedback on our article. We appreciate your recognition of our framework and the approach taken to assess the impact of celiac disease on the quality of life in pediatric patients. Your point about the crucial role of parents in initiating the diagnostic process is particularly important, as their observations often lead to early medical intervention.
We also acknowledge the significance of specialized nutritional teams in second- and third-level hospitals, as their expertise extends beyond oncology patients to include pediatric cases, ensuring comprehensive dietary management. We appreciate your reference to the study (doi.org/10.3390/nu17010188), which we have reviewed and incorporated into our discussion to further support our findings.
Regarding the administration of the questionnaire, we agree that the involvement of parents or caregivers could influence responses. Identifying a third party, such as a hospital social worker, as an objective evaluator is an excellent suggestion that could enhance the accuracy of reported outcomes. – we discussed this in limitation of study
Finally, we are pleased that our discussion, conclusions, and bibliography align with the paper’s objectives and scientific standards. Your comments have been invaluable in refining our work, and we sincerely appreciate your thoughtful review.
Round 2
Reviewer 1 Report
Comments and Suggestions for Authors
I find the revised manuscript substantially improved and believe it now meets the standards for publication. I recommend acceptance of the manuscript in its current form.